Optimizing total RNA extraction method for human and mice samples

Zeng Yumei 1
Tang Xiaoxue 2
Chen Jinwen 3
Kang Xi 1
Bai Dazhang baidazhang@126.com 1 2
1 Department of Neurology, Affiliated Hospital of North Sichuan Medical College , Nanchong , Sichuan , China
2 Institute of Neurological Diseases, Affiliated Hospital of North Sichuan Medical College , Nanchong , China
3 Department of Clinical Laboratory, Affiliated Hospital of North Sichuan Medical College , Nanchong , China
Sotelo-Mundo Rogerio
Electronic publication date: 2024 Sep 26
Publication date: 2024
Volume: 12
Electronic Location ID: e18072
Received 2024 May 22; Accepted 2024 Aug 19
Copyright: ©2024 Zeng et al.
Copyright year: 2024
Copyright holder: Zeng et al.
License: This is an open access article distributed under the terms of the Creative Commons Attribution License, which permits unrestricted use, distribution, reproduction and adaptation in any medium and for any purpose provided that it is properly attributed. For attribution, the original author(s), title, publication source (PeerJ) and either DOI or URL of the article must be cited.
License URL: https://creativecommons.org/licenses/by/4.0/

Keywords: Total RNA extraction method, TRIzol reagent, The TRIzol method, GITC-T method, SDS-T method

Funding: The Doctoral Research Foundation of North Sichuan Medical College CBY21-QD17 The City-School Science and Technology Strategic Cooperation Project of Nanchong 22SXQT0032 The Natural Science Foundation of Sichuan Province, Sichuan Science and Technology Program 2023NSFSC0709 This work was supported by the Doctoral Research Foundation of North Sichuan Medical College (CBY21-QD17); the City-School Science and Technology Strategic Cooperation Project of Nanchong (22SXQT0032); the Natural Science Foundation of Sichuan Province, Sichuan Science and Technology Program (2023NSFSC0709). The funders had no role in study design, data collection and analysis, decision to publish, or preparation of the manuscript.

==============================
Background

Extracting high-quality total RNA is pivotal for advanced RNA molecular studies, such as Next-generation sequencing and expression microarrays where RNA is hybridized. Despite the development of numerous extraction methods in recent decades, like the cetyl-trimethyl ammonium bromide (CTAB) and the traditional TRIzol reagent methods, their complexity and high costs often impede their application in small-scale laboratories. Therefore, a practical and economical method for RNA extraction that maintains high standards of efficiency and quality needs to be provided to optimize RNA extraction from human and mice tissues.

Method

This study proposes enhancements to the TRIzol method by incorporating guanidine isothiocyanate (GITC-T method) and sodium dodecyl sulfate (SDS-T method). We evaluated the effectiveness of these modified methods compared to the TRIzol method using a micro-volume UV-visible spectrophotometer, electrophoresis, q-PCR, RNA-Seq, and whole transcriptome sequencing.

Result

The micro-volume UV-visible spectrophotometer, electrophoresis, and RNA-Seq demonstrated that the GITC-T method yielded RNA with higher yields, integrity, and purity, while the consistency in RNA quality between the two methods was confirmed. Taking mouse cerebral cortex tissue as a sample, the yield of total RNA extracted by the GITC-T method was 1,959.06 ± 49.68 ng/mg, while the yield of total RNA extracted by the TRIzol method was 1,673.08 ± 86.39 ng/mg. At the same time, the OD260/280 of the total RNA samples extracted by the GITC-T method was 2.03 ± 0.012, and the OD260/230 was 2.17 ± 0.031, while the OD260/280 of the total RNA samples extracted by the TRIzol method was 2.013 ± 0.041 and the OD260/230 was 2.11 ± 0.062. Furthermore, q-PCR indicated that the GITC-T method achieved higher yields, purity, and greater transcript abundance of total RNA from the same types of animal samples than the TRIzol method.

Conclusion

The GITC-T method not only yields higher purity and quantity of RNA but also reduces reagent consumption and overall costs, thereby presenting a more feasible option for small-scale laboratory settings.

Introduction

In recent years, ribonucleic acid (RNA)-based research methodologies have advanced significantly, encompassing techniques like RNA hybridization, real-time fluorescent quantitative polymerase chain reaction (q-PCR), RNA sequencing (RNA-Seq), and whole transcriptome sequencing. These methods have garnered considerable interest and application across various domains, including public health (Torii, Furumai & Katayama, 2021; Hoffman et al., 2022), clinical diagnostics (Yüce, Filiztekin & Özkaya, 2021; Gagliardi et al., 2021), and life sciences (Clark et al., 2019). Despite the evolution of RNA research techniques, small-scale domestic laboratories often encounter obstacles in adopting these advanced methods due to technical complexities and resource limitations. The commonly employed methods for RNA extraction, such as phenol-chloroform extraction (Dimke et al., 2021; Hoffman et al., 2022), density gradient centrifugation (Weis, Schnell & Egert, 2020), TRIzol (Ma et al., 2010), and various commercial kits including spin column (Biró et al., 2019), silica column (Yang et al., 2017), and magnetic bead (He et al., 2017; Klein et al., 2020), present their challenges. Traditional techniques like phenol-chloroform extraction and density gradient centrifugation are labor-intensive and complex, hindering widespread adoption.

Conversely, the traditional TRIzol reagent method (the TRIzol method), centrifugal column extraction, silica gel column extraction, and magnetic bead extraction are easy to use but costly and not suitable for large-scale use in small-scale laboratories (Brown et al., 2018; Scholes & Lewis, 2020; Schactler et al., 2023). To obtain an easy-to-operate and inexpensive RNA extraction method, researchers have been continuously trying to improve the RNA extraction reagents or extraction methods, with most of the improved methods based on the TRIzol reagent (Duy et al., 2015; Gandhi, O’Brien & Yadav, 2020; Schactler et al., 2023). However, these modified methods either failed to reduce the cost of total RNA extraction experiments or increased the complexity of the total RNA extraction process. Therefore, we propose changing the TRIzol method to create a simple and inexpensive approach for total RNA extraction.

The conventional TRIzol reagent is recognized for its stability and efficiency in extracting total RNA (Kao et al., 2023). However, its relatively high cost can lead to attempts to minimize reagent use during experiments, potentially resulting in organic residue contamination and impacting subsequent molecular experiments. Phenol and guanidine isothiocyanate (GITC) are the main components of the traditional TRIzol reagent, while GITC is also a cost-effective auxiliary reagent increasingly employed in improved RNA extraction methods. The GITC, with strong protein denaturing capabilities, aids in cell membrane disruption and disrupts protein-nucleic acid interactions, effectively inactivating ribonucleases (RNases) in cells (Ghawana et al., 2011). This action is crucial for releasing and preserving intact RNA. Studies have shown that GITC not only inhibits RNase activity but also plays a role in the phase separation of nucleic acids, which is adjustable through concentration modification. Sodium dodecyl sulfate (SDS), an effective anionic surfactant in RNA extraction, assists in disrupting cell and nuclear membranes and emulsifying lipids. Its role is vital in denaturing proteins and detaching them from RNA, facilitating the release and preservation of RNA (Barbier et al., 2019; Vennapusa et al., 2020). Since GITC and SDS are relatively inexpensive and low-toxicity reagents, they are often used to improve the extraction method of total RNA.

In this study, we introduced the addition of GITC (GITC-T method) and SDS (SDS-T method) to the commercial TRIzol reagent process for extracting total RNA from human and mouse samples. The primary aim was to reduce the volume of the TRIzol reagent required, thereby decreasing experimental costs while still obtaining RNA products of similar or enhanced quality. By modifying the conventional process with these additions, we aimed to provide a straightforward, cost-effective, and universally applicable method for total RNA extraction from human and mouse samples, specifically tailored to meet the needs of small-scale laboratories facing financial constraints.

Material and Methods

Experimental materials

C57BL/6J Mice: Obtained from Changzhou Cavins Laboratory Animal Co., Ltd and these mice were fed in the Laboratory Animal Center of North Sichuan Medical College on a normal diet, weighing between 20–25 g, and were aged 6-8 weeks. All animal experimental procedures were approved by the Animal Research and Ethics Committee of North Sichuan Medical College (approval number NSMC(A)2021(114)). Mice were anesthetized with 3% isoflurane and executed by neck dissection.

Human glioblastoma cells (U87-MG): Sourced from Wuhan Procell Life Science & Technology Co., Ltd (catalog no: CL-0238). Human cervical carcinoma cells (Hela S3): acquired from Sichuan Bio Biotechnology Co., Ltd (catalog no: B26087).

Blood samples: Gathered from healthy adults at the Affiliated Hospital of North Sichuan Medical College or ourselves. The procedures for human blood sample collection were approved by the Ethics Committee of the Affiliated Hospital of North Sichuan Medical College (file number: 2023ER372-1).

Experimental reagents and instruments

Reagents

Chemicals: Included but not limited to, in the study, GITC, SDS, chloroform, isopropanol (IPA), 75% ethanol, agarose, aminomethane (Tris), Ethylenediamine tetraacetic acid (EDTA), and anhydrous acetic acid.

Commercial reagents: Included but not limited to the TRIzol reagent, Eagle’s Basic Medium (BME), F-12K medium, and fetal bovine serum (FBS), procured from Thermo Fisher Scientific (Waltham, MA, USA). Additional reagents included diethyl pyrocarbonate-treated water (DEPC water), trypsin-EDTA solution, penicillin-streptomycin solution, and Dulbecco’s phosphate-buffered saline (DPBS) from Ranjco Technology Co., Ltd. The reverse transcription kit (R-T Kit) and real-time fluorescence quantitative PCR kit (q-PCR Kit) were acquired from TaKaRa (Shiga, Japan), with q-PCR primer pairs synthesized by Shanghai Sangong Biotechnology Co., Ltd (Shanghai, China). More details about the reagents are referenced in Table S1.

Instruments

Homogenization was performed using a Handheld homogenizer (Tengen Biochemical Technology Co. Beijing, China, Ltd OSE-Y50). Spectrophotometer: NanoDrop™ One Micro-volume UV-Vis Spectrophotometer (Thermo Fisher Scientific, Inc., Waltham, MA, USA). Electrophoresis: Mini ReadySub-Cell GT Horizontal Electrophoresis System (Bio-Rad Laboratories, Inc., Hercules, CA, USA). Gel Documentation: Gel documentation imaging system (GenoSens 2000, Clinx Science Instruments Co., Ltd). PCR Analysis: CFX Opus 96 Real-Time PCR System (Bio-Rad Laboratories, Inc., Hercules, CA, USA). More information on instruments is summarized in Table S2.

Experimental Methods

Preparation of tissue samples

C57BL/6J mice who are about eight weeks old were humanely euthanized under deep anesthesia using 3% isoflurane gas, adhering to approved ethical guidelines. Brain tissues were quickly extracted using humane methods. The cerebral cortex was separated on ice in pre-chilled disposable cell culture dishes to maintain tissue integrity. The tissues were then gently homogenized, aliquoted into sterile, enzyme-free 1.5 ml Eppendorf centrifuge tubes (EP tubes), and accurately weighed. These prepared cerebral cortex samples were preserved on ice to ensure freshness until further processing.

Preparation of cell samples

U87-MG and Hela S3 were cultured in BME and F-12K medium, respectively, both supplemented with 10% FBS and 1% penicillin-streptomycin solution. The cells were incubated at 37 °C in a 5% CO2 atmosphere. At the exponential growth phase, the original cell culture was discarded. Cells were rinsed with 1 mL of Dulbecco’s phosphate-buffered saline (DPBS), followed by discarding the DPBS. Subsequently, 1 mL of trypsin-EDTA solution was added for cell detachment and incubated at 37 °C for 3 min. The trypsinization was stopped by adding 2 mL of the respective complete medium. The cells were resuspended through gentle pipetting and transferred to a 15 mL centrifuge tube for centrifugation at 100 ×g for 5 min. The supernatant was discarded, and the cells were resuspended in 3 mL of DPBS, equally distributed into 3 sterile, enzyme-free 1.5 mL EP tubes, and centrifuged at 200 ×g for 5 min at 4 °C. The supernatant was discarded, and the cell pellets were kept on ice.

Preparation of blood samples

Blood samples from healthy adults were collected into vacuum blood collection tubes with purple caps, indicating the presence of EDTA as an anticoagulant. The samples were mixed thoroughly by gentle inversion and aliquoted into three sterile, enzyme-free 1.5 ml EP tubes, each receiving 200 µl of blood. These samples were then stored temporarily at 4 °C for backup.

Total RNA extraction using the TRIzol method

Total RNA was extracted from blood, cell, and tissue samples, including mouse cerebral cortex tissues, following the guidelines of the manufacturer manual, which is modified slightly by us to elevate the yield of total RNA. The procedure is summarized as follows:

Lysis: Initially, 500 µl of the TRIzol reagent was added to each sample tube, and the tissues were homogenized using a handheld homogenizer. An additional 500 µl of TRIzol was then added to ensure complete lysis.

Phase separation: Following 5-min incubation at room temperature, 200 µl of chloroform was added. The tubes were vigorously shaken to form a pink emulsion and allowed to hold for 3 min at room temperature before undergoing centrifugation at 12,000 ×g for 15 min at 4 °C. The aqueous phase was meticulously transferred to a new tube to avoid protein contamination.

RNA precipitation: An equal volume of IPA was mixed well with the supernatant, and the samples were left to precipitate at −20 °C overnight. The following day, the supernatant was discarded after centrifugation at 4 °C for 15 min at 12,000 ×g.

Washing: The RNA pellet underwent washing with 1 ml of 75% ethanol by centrifuging at 7,500 × g for 5 min at 4 °C, followed by a second spin at 12,000 × g for 5 min to ensure thorough washing.

Drying and dissolving: The ethanol was discarded, and the RNA pellet was air-dried with open caps for 5-10 min. Subsequently, the dried RNA was redissolution in 20 µl of DEPC water, ensuring complete dissolution before storage at −80 °C.

Optimization of the amount of GITC and SDS additions

This method was explicitly applied to mouse cerebral cortex tissue samples as outlined below:

Lysis: Initially, 200 µl of 3 mol/L (M), 4 M, and 5 M for GITC and 5%, 10%, and 15% for SDS solution were added separately to the prepared sample tube, followed by homogenization with a handheld homogenizer. Subsequently, 800 µl of TRIzol reagent was added to the EP tube and thoroughly mixed. The remaining steps are the same as those in the TRIzol method.

Total RNA extraction using the GITC-T method

This method was applied to mouse cerebral cortex tissue samples as follows:

Lysis: Initially, 100 µl of 20% SDS solution was added to the prepared sample tubes and homogenized with a handheld homogenizer. Subsequently, 800 µl of TRIzol reagent and 100 µl of GITC solution were added. Mix the lysate well and place on ice for 15 min. The remaining steps are the same as those in the TRIzol method.

Total RNA extraction using the SDS-T method

This method was explicitly applied to mouse cerebral cortex tissue samples as outlined below:

Lysis: Initially, 200 µl of 10% SDS solution was added to the prepared sample tube, followed by homogenization with a handheld homogenizer. Subsequently, 800 µl of TRIzol reagent was added to the EP tube and thoroughly mixed, then placed on ice for 15 min. The remaining steps are the same as those in the TRIzol method, and the procedure of the three animal sample total RNA extraction methods is summarized in Table 1.

Table 1 Protocols for extraction of total RNA from animal samples.

Component	Protocol	
	TRIzol	SDS-T	GITC-T	
10% SDS (µl)		200		
20% SDS (µl)			100	
TRIzol (µl)	1,000	800	800	
GITC (µl)			100	
Incubation condition	room temperature	ice bath	ice bath	
Incubation	5 min	15 min	15 min	
Chloroform(µl)	200	200	200	
Incubation	3 min	3 min	3 min	
Centrifugal conditions	4 °C 12,000 × g 15 min	4 °C 12,000 × g 15 min	4 °C 12,000 × g 15 min	
Isopropanol	Equal to RNA supernatant  volume	Equal to RNA supernatant  volume	Equal to RNA supernatant  volume	
Precipitate the RNA	−20 °C overnight	−20 °C overnight	−20 °C overnight	
Centrifugal conditions	4 °C 12,000 × g 15 min	4 °C 12,000 × g 15 min	4 °C 12,000 × g 15 min	
75% ethanol (µl)	1,000	1,000	1,000	
Centrifugal conditions	4 °C 7,500 × g 5 min, then 12,000 × g 5 min	4 °C 7,500 × g 5 min, then 12,000 × g 5 min	4 °C 7,500 × g 5 min, then 12,000 × g 5 min	
Dissolution	20µl DEPC water	20µl DEPC water	20µl DEPC water	

Total RNA yield and purity assay

The yield and purity of total RNA extracted from human and mouse samples were assessed using Thermo’s NanoDrop™ One Micro-volume UV-Vis Spectrophotometer. The indicators of RNA purity focus on the absorbance of the OD260/OD280 and OD260/OD230 ratios. The procedure is outlined as follows:

Instrument preparation: The spectrophotometer was meticulously cleaned before the testing to ensure accurate purity assessments.

Baseline calibration: DEPC water served to establish a blank sample for calibrating the absorbance baseline specific to the RNA solvent.

Sample measurement: Take one microliter of each RNA sample for concentration and purity detection. The purity of RNA samples is mainly reflected in the ratio of OD260/OD280 and OD260/OD230. Each sample underwent three separate measurements.

Data analysis: The mean value and standard error (mean   ±  SEM) were calculated from the nine data points to assess the efficiency, purity levels, and reproducibility of the different RNA extraction methods.

Total RNA integrity assay

The integrity of the total RNA samples was evaluated using agarose gel electrophoresis, following these steps:

Each RNA sample was electrophoresed using an agarose gel at a concentration of 1% containing an ethidium bromide substitute. The parameters of agarose gel electrophoresis are voltage 120 V and time 45 min. After electrophoresis, electropherograms were acquired using the Gel Documentation Imaging System, and 28S and 18S band signals were acquired using Image J (version 1.8.0) software to assess the integrity of each RNA sample.

Abundance detection of transcripts in total RNA samples

The quantification of specific gene transcripts within total RNA samples involved several key steps:

Primer design: Primers for the human housekeeping gene Glyceraldehyde phosphate dehydrogenase (GAPDH, Gene ID: 2597) and the long non-coding gene PU.1 induced regulator of S100A8 and S100A9 alarmin transcription 1 (PIRAT1, Gene ID: 101929559) were identified using the “Pick Primers” tool on the NCBI website (https://www.ncbi.nlm.nih.gov). The selected primers span at least one intron to ensure specificity for cDNA. Sequence information of primer pairs is shown in Table S3.

cDNA Synthesis: Genomic DNA (gDNA) is removed by DNase, which is in a reverse transcript kit (#RR047A; TaKaRa, Shiga, Japan). Then 1000 ng of total RNA was reverse transcribed into complementary DNA (cDNA) following the protocol provided with the reverse transcription kit.

q-PCR Setup: The q-PCR reaction mix was prepared according to the quantitative PCR kit’s instructions (#RR820A; TaKaRa, Shiga, Japan). PCR amplification was conducted on the CFX96 Real-Time PCR System, with a total reaction volume of 25 µL. The cycling conditions were as follows: an initial pre-denaturation at 95 °C for 30 s, followed by 40 cycles of denaturation at 95 °C for 5 s, and annealing/extension at 60 °C for 30 s. The melting curves of the q-PCR products were analyzed from 65 °C to 95 °C.

Data analysis: The relative abundance of the target gene transcript in the RNA sample was determined by analyzing the threshold cycle (Ct) of the q-PCR reaction to evaluate which total RNA sample had more starting copies of the target gene transcript (Livak & Schmittgen, 2001; Schmittgen & Livak, 2008).

Transcriptome and whole transcriptome sequencing

RNA samples, including those extracted from human whole blood using both the TRIzol method and the GITC-T method, as well as RNA from Hela S3 and U87-MG cell lines, were forwarded to Wuhan Gene Read Biotechnology Co., Ltd (Wuhan, China). for comprehensive RNA sequencing analysis.

The RNA-Seq samples were constructed using the VAHTS® Universal V8 RNA-seq Library Prep Kit for Illumina (#NR605; Vazyme, Nanjing, China). Whole transcriptome sequencing samples require the construction of two sequencing libraries: first, the VAHTS® Small RNA Library Prep Kit for Illumina V2 (#NR811; Vazyme, Nanjing, China) is used to construct small RNA sequencing libraries. Second, ribosomal RNA was removed using the Ribo-MagOff rRNA Depletion Kit (#N420; Vazyme, Nanjing, China), and then the VAHTS® Universal V8 RNA-seq Library Prep Kit for Illumina (#NR605; Vazyme, Nanjing, China) was used to construct lncRNA-sequencing libraries. Finally, all libraries were sequenced using the Illumina NovaSeq 6000 platform.

Statistical analysis

Statistical analysis was performed for each total RNA extraction method, involving at least three separate extraction trials. These trials encompassed a variety of animal samples, including mouse cerebral cortex tissue, human tumor cells, and human blood samples. Two-sided t-tests were utilized to compare data between two distinct groups, evaluating statistical differences. In cases of multiple group comparisons, either one-way or two-way ANOVA was employed, complemented by Tukey’s multiple comparisons test to ascertain statistical significance. The findings were presented as mean  ±  SEM, with GraphPad Prism 8 software used for computation and visualization.

Results

Higher yield of total RNA from animal samples extracted by the GITC-T method

The study examined the effect of adding GITC and SDS on the total RNA yield from animal samples, particularly mouse cerebral cortex tissues. Concentration gradients of 3 mol/L (M), 4 M, and 5 M for GITC and 5%, 10%, and 15% for SDS were tested to determine the optimal concentrations. RNA sample concentrations detected by NanoDrop™ One (Table 2) and their statistical bar plots (Figs. 1A and 1B) showed that the yield of total RNA increased with increasing GITC concentration. At the same time, the yield of total RNA peaked at 10% SDS, and too high or too low SDS concentrations reduced the yield of RNA. This is also shown for the electrophoresis patterns (Figs. 1C and 1D) and their statistical bar plots (Figs. 1E and 1F) of RNA samples.

Upon identifying the optimal concentrations of GITC (5 M) and SDS (10%), the study compared total RNA yields from the same animal samples using three extraction methods: the TRIzol method, GITC-T, and SDS-T methods. NanoDropTM One quantified the yields, which were normalized to the unit weight of the mouse cerebral cortex tissues. The GITC-T exhibited the highest RNA yield, surpassing that of the TRIzol method, while the SDS-T yielded the least. Detailed outcomes are provided in Table 3 and Fig. 1G.

Higher purity of total RNA from animal samples extracted by GITC-T method

The purity of total RNA extracted from animal samples using the GITC-T method was evaluated by measuring OD260/OD280 and OD260/OD230 ratios with NanoDropTM One. An OD260/OD280 ratio below 1.9 suggests protein contamination, while a ratio above 2.1 indicates potential DNA contamination or RNA degradation (Desjardins & Conklin, 2010). Similarly, an OD260/OD230 ratio below 2.0 indicates salt contamination, whereas a ratio above 2.0 signifies high-purity RNA without salt contamination (Ahlberg, Jenmalm & Tingö, 2021). According to the results presented in Table 4 and Fig. 2, RNA samples extracted via the GITC-T method exhibited the most minor contamination by proteins and salt ions, surpassing those obtained through the TRIzol method. Conversely, the SDS-T method showed the highest levels of protein and salt ion contamination. Consequently, further comparative experiments focused on the GITC-T and the TRIzol methods to assess their efficacy in RNA extraction.

Table 2 Effect of different GITC or SDS dosages on total RNA yield.

Concentration of additives	Weight of cortex  sample (mg)	Volume of dissolved DEPC water (µl)	RNA concentration (ng/µl)	RNA extraction rate  (ng/mg)	
GITC (3 mol/L)	7.46 ± 0.25	20	401.10 ± 3.26	1,076.76 ± 44.93	
GITC (4 mol/L)	7.35 ± 2.87	20	404.17 ± 173.45	1,084.61 ± 53.68	
GITC (5 mol/L)	5.77 ± 1.15	20	327.03 ± 63.55	1,135.89 ± 26.59	
SDS (5%)	74.37 ± 10.53	20	3,977.50 ± 619.08	1,067.30 ± 21.36	
SDS (10%)	80.00 ± 11.57	20	4,127.23 ± 693.45	1,083.45 ± 19.10	
SDS (15%)	60.00 ± 2.94	20	2,895.10 ± 352.64	1,012.34 ± 70.28	

Figure 1 Effect of different additives on RNA yield using mice tissue samples.

(A) The statistical results of Table 2 showed that the highest RNA yield was obtained when the concentration of GITC was 5 M (* p < 0.05; ** p < 0.01; ns: no significant). (B) The statistical results of Table 2 showed that the yield of RNA was the highest when the concentration of SDS was 10% (* p < 0.05; ns: no significant). (C) The results of agarose gel electrophoresis showed that the signal intensity of RNA increased with the increase of GITC concentration, but the integrity of total RNA was basically the same. (D) The results of agarose gel electrophoresis showed that the signal intensity of RNA was strongest at 10% SDS concentration, and the integrity of total RNA was basically consistent. (E) and (F) counted the signal intensities of the 28S and 18S bands in electrophoresis plots (C) and (D), respectively(** p < 0.01; *** p < 0.001; **** p < 0.0001; ns: no significant). (G) Statistical results of Table 3 showed that the GITC-T method combining 5 M GITC and 10% SDS could obtain the best RNA yield (**** p < 0.0001).

The q-PCR results (Fig. S2) show that for total RNA samples extracted from human blood using the TRIzol method, the Ct values are significantly lower in the group without gDNA removal compared to the group with gDNA removal. That indicates a high amount of residual gDNA in the total RNA samples. Conversely, for total RNA samples extracted using the GITC-T method, the Ct value difference between the two groups is much smaller, suggesting less residual gDNA in these samples. Thus, it concluded that the total RNA samples extracted using the GITC-T method have less residual gDNA than those extracted using the TRIzol method, implying higher total RNA purity.

Higher integrity of total RNA extracted by the GITC-T method

The integrity of total RNA extracted from animal tissues and cells was notably higher with the GITC-T method. Typically, animal cells yield three primary RNA types—28S, 18S, and 5S—distinguishable by agarose gel electrophoresis. RNA integrity is inferred from the 28S to 18S band intensity ratio, with the ideal ratio being approximately two-fold higher for the 28S bands. Figs. 1C, 1D, and Fig. S1 illustrate that both the GITC-T and the TRIzol methods produced clear bands for all three RNA types, indicating good integrity. Furthermore, the integrity assessment extended to U87-MG cell samples processed by both methods, with sequencing pre-sequencing quality tests conducted by Wuhan Gene Read Biotechnology Co., Ltd using the Bioptic Qsep 100 bioanalyzer. The findings, depicted in Fig. 3A, showed that the GITC-T method yielded a larger area under the 28S peak and a higher 28S/18S ratio compared to the TRIzol method, affirming the superior integrity of RNA extracted via the GITC-T method.

In addition, total RNA samples extracted by the GITC-T and TRIzol methods were subjected to denaturing agarose gel electrophoresis after RNA-Seq and stored in Wuhan Gene Read Biotechnology Co., Ltd. The results of electrophoresis are shown in Fig. S3, and the 28S, 18S, and 5S bands of each RNA sample are clearly visible, and the brightness of the 28S band significantly exceeds the brightness of the 18S band (about 2-fold), confirming that the RNA samples obtained by the total RNA extraction methods of the two animal samples are of good integrity. At the same time, the total RNA extracted by the GITC-T method from the same sample had a higher signal intensity than that extracted by the TRIzol method.

Table 3 Yield of total RNA from different extraction methods.

Protocol	Weight of cortex sample (mg)	Volume of the aqueous  phase (µl)	RNA concentration  (ng/µl)	RNA extraction rate  (ng/mg)	
TRIzol	68.23 ± 13.92	20	5,724.37 ± 1,247.47	1,673.08 ± 86.39	
GITC-T	60.50 ± 8.32	20	5,946.73 ± 950.18	1,959.06 ± 49.68	
SDS-T	78.50 ± 12.03	20	5,159.53 ± 1,146.95	1,302.19 ± 89.49	

Table 4 Purity of total RNA from different extraction methods.

protocol	RNA extraction rate (ng/mg)	OD 260/280	OD 260/230	
TRIzol	1,673.08 ± 86.39	2.013 ± 0.041	2.11 ± 0.062	
GITC-T	1,959.06 ± 49.68	2.03 ± 0.012	2.17 ± 0.031	
SDS-T	1,302.19 ± 89.49	2.02 ± 0.036	2.07 ± 0.101	

Figure 2 Effect of different extraction methods on the purity of total RNA using mice tissue samples.

(A) The bar plot drawn from the data in Table 4 shows that the total RNA extracted by the GITC-T method has the optimal OD260 to OD280 ratio (*** p < 0.001; ns: no significant). (B) Total RNA extracted by the GITC-T method has the best OD260 to OD230 ratio, although it did not present a significant difference (ns: no significant).

Figure 3 The integrity and consistency of RNA obtained by two total RNA extraction methods using blood, U87-MG and Hela S3 cell samples.

(A) Results of U87-MG cell of RNA sample integrity quality-control testing provided by Wuhan Gene Read Biotechnology Co., LTD, using the Bioptic Qsep 100 bioanalyzer. The integrity of total RNA obtained from U87-MG cells extracted by the GITC-T method was higher compared to the TRIzol method. The left part is presented as the chromatogram, and the right part is the corresponding detection data. (B) Comparison of consistency between RNA from different types of samples by both methods. As shown in the Fig, the RNA consistency between different samples is poor, but the RNA consistency of the same sample obtained by different extraction methods is very good. Thicker color of the ruler indicates better sample consistency.

There is no difference between the total RNA extracted by the GITC-T method and the TRIzol method

The study compared total RNA extracted from human blood, Hela S3 cells, and U87-MG cells using both the GITC-T and the TRIzol methods, with samples sent to Wuhan Gene Read Biotechnology Co., Ltd for RNA-seq sequencing or whole transcriptome sequencing. Quality control results, presented in Fig. 3B, demonstrated consistent RNA quality between the two extraction methods across various sample types, suggesting no significant difference in the total RNA extracted by either method. Furthermore, RNA-Seq analysis of cells revealed comparable RNA sequence compositions between samples processed with the TRIzol method (Fig. 4A) and those with the GITC-T method (Fig. 4B), reinforcing the conclusion that the two methods yield essentially equivalent total RNA in terms of quality and composition.

Figure 4 RNA sequence analysis based on RNA-seq data using blood, U87-MG and Hela S3 cell samples.

The RNA sequences of the TRIzol method (A) and GITC-T method (B) did not differ in terms of composition, but there were differences in the yield ratio of each component. Compared with the TRIzol method (A), the total RNA samples obtained by the GITC-T method (B) had a larger proportion of introns, suggesting that the GITC-T method (B) had a better nuclear membrane cleavage effect and could obtain more precursor mRNA.

The total RNA transcript abundance of animal samples extracted by the GITC-T method was higher

The study compared transcript abundance in total RNA extracted from animal samples using the GITC-T method versus the TRIzol method. Quantitative amplification was performed following the q-PCR kit instructions, with primer pairs listed in Table S3 and the reaction system detailed in Table S4. The melting curve for the q-PCR primer products revealed a single peak for the GAPDH and PIRAT1 primers, indicating high primer specificity (Fig. 5A). To eliminate the interference of reagents and gDNA residues on the q-PCR results of total RNA samples extracted by different methods, we set three q-PCR groups: no template group, no gDNA removal group, and gDNA removal group. The q-PCR amplification curves demonstrated earlier peaks for samples extracted with the GITC-T method, suggesting a higher number of transcript copies for the GAPDH and PIRAT1 genes compared to those extracted by the TRIzol method (Fig. 5B). Statistical analysis confirmed that the q-PCR Ct values for the GITC-T method were lower, indicating a significant difference in GAPDH and PIRAT1 transcript copy numbers between the two extraction methods (Fig. 5C).

Figure 5 Quantification of RNA transcript obtained by different methods of total RNA extraction using blood samples.

(A) Melting curves of the amplified products of two pairs of q-PCR primers, the amplification products of the GAPDH transcript on the left and those of the PIRAT1 transcript on the right. (B) GAPDH and PIRAT1 pairs of q-PCR primers performed q-PCR on the total RNA samples of human whole blood extracted by the TRIzol and GITC-T methods. During the experiment, three groups were set up: no template, preserving gDNA, and erasing gDNA. The presence of residual gDNA in the total RNA sample would make the fluorescent signal of the q-PCR reaction reach the detection threshold earlier. (C) According to the q-PCR cycle threshold (Ct) values of GAPDH and PIRAT1 transcripts in (B), the RNA samples extracted by GITC-T method have smaller Ct values. This indicated that the content of GAPDH and PIRAT1 transcripts in RNA extracted by GITC-T method was higher than that by the TRIzol method (** p < 0.01; **** p < 0.001).

The total RNA extracted from animal samples by the GITC-T method was superior to the TRIzol method

The GITC-T method for extracting total RNA from animal samples demonstrated superiority over the TRIzol method across multiple metrics. While the consistency in RNA quality between the two methods was confirmed (Fig. 3B), the GITC-T method yielded RNA with higher integrity (Fig. 3A) and greater transcript abundance (Figs. 5B and 5C). Furthermore, comparisons revealed that the GITC-T method achieved higher yields (Table 3, Fig. 1G) and purity (Table 4, Fig. 2, Tables S5) of total RNA from the same types of animal samples than the TRIzol method. These advantages highlight the GITC-T method’s overall superiority in extracting total RNA, offering the additional benefit of reducing experimental costs by minimizing the use of the traditional TRIzol reagent.

Discussion

RNA, a complex and multifaceted biomolecule, remains a pivotal subject of study in life sciences and medicine (Roszkowski & Mansuy, 2021). Essential for a range of analyses such as q-PCR, RNA-seq, and whole transcriptome sequencing, high-quality total RNA samples are fundamental (Roszkowski & Mansuy, 2021; Dandare et al., 2022; Zhao et al., 2023). While the cetyl-trimethyl ammonium bromide (CTAB) method is traditionally employed for extracting total RNA from plant samples (Sasi et al., 2023; Mainkar et al., 2023), the TRIzol method is the standard for animal samples (Chomczynski & Sacchi, 2006). Despite advancements in technology enhancing the diversity of RNA extraction methods for animal samples, surpassing the efficacy of the TRIzol method proves challenging. Techniques like spin column extraction, though streamlining the process, often result in lower RNA yields (Roos-van Groningen et al., 2004; Yang et al., 2017). Similarly, magnetic bead extraction achieves high purity but is deterred by higher costs (Butcher et al., 2014; Adams et al., 2015). This study aims to identify a total RNA extraction method from animal samples that offers both cost-efficiency and RNA quality comparable or superior to the TRIzol method. Building on previous research to enhance animal sample RNA extraction methods (Rodgers et al., 2022; Faraldi et al., 2022; Avramov et al., 2024), the focus is on refining the TRIzol method to balance cost-effectiveness with high-quality RNA yield.

In this research, GITC, a potent protein denaturant, and SDS, an anionic surfactant, were employed for their cost-effectiveness and efficacy in disrupting cell membranes to release and safeguard RNA during total RNA extraction from animal samples (Singer & Tjeerdema, 1993; Ogram et al., 1995; Otzen et al., 2022). Despite their everyday use, the study encountered several challenges:

1.   SDS precipitation: SDS solution-containing sample homogenates precipitated SDS crystals when cooled on ice, leading to a diminished RNA yield. This indicates the need for careful temperature management when using SDS in RNA extraction processes.

2.   Inadequacy for serum or plasma samples: Neither 200 µL of serum nor plasma provided sufficient RNA for q-PCR analysis. This outcome suggests that the TRIzol method and its modifications may not be optimal for extracting RNA from serum or plasma samples. This finding differs somewhat from the results of Chen et al. (2023).

3.   Heparin anticoagulant interfered with the q-PCR assay: While the GITC-T, TRIzol, and SDS-T methods successfully extracted RNA from heparin-anticoagulated whole blood samples, the resultant RNA failed to produce amplification products in q-PCR experiments. Conversely, RNA extracted from EDTA-anticoagulated samples did not face this issue, indicating that heparin may interfere with RNA quality or q-PCR reactions, rendering heparin-anticoagulated samples unsuitable for such analyses.

4.   Although the GITC and SDS can be used together or separately to improve the traditional TRIzol method when extracting RNA by the GITC-T method, SDS and GITC should not be added successively. The TRIzol reagent must be added after SDS to act as a buffer, or crystals will precipitate when they meet directly. In the future, we can change the order of adding the three reagents, such as adding GITC first, then TRIzol, and finally SDS, to see if we can achieve similar or better total RNA extraction efficiency from animal samples.

5.   GITC concentration optimization: The study experimented with three GITC concentration gradients—3 M, 4 M, and 5 M—and observed that RNA yield increased with GITC concentration. This suggests further increasing the GITC concentration or reducing the TRIzol reagent volume might enhance RNA yield and overall experimental outcomes. Of course, this speculation still needs further experimental verification.

Denaturing gel electrophoresis offers precise measurement of RNA molecular weights and integrity assessment, yet standard agarose gel electrophoresis is favored for its convenience and quickness in evaluating RNA integrity (Figs. S1 and S3). Although the theoretical ideal for the 28S/18S ribosomal RNA band brightness ratio is 2.7:1, a 2:1 ratio is commonly accepted to indicate good RNA integrity. In practice, clear visibility of 28S, 18S, and 5S RNA bands with a 28S:18S ratio exceeding 1.0 is sufficient for most experimental needs. The GITC-T and the TRIzol methods can yield RNA of satisfactory integrity (Fig. 3A and Fig. S3). Consistency analyses of RNA-seq data from samples extracted using either method revealed high correlation coefficients, nearing 1.0 (Fig. 3B), suggesting that the total RNA quality is highly similar regardless of the extraction method. This similarity indicates that variations in RNA samples are more likely attributed to differences in sample types rather than the extraction techniques employed. Therefore, substituting the TRIzol method with the GITC-T method, which offers higher yields, is unlikely to introduce biases in experimental outcomes.

Moreover, compositional analysis of RNA-seq data revealed no significant differences in the sequence component composition between the GITC-T and the TRIzol methods (Fig. 4). However, there was a notable increase in the percentage of intronic sequences in RNA samples extracted with the GITC-T method. This suggests the presence of a higher proportion of precursor mRNAs, implying that the GITC solution may more effectively disrupt cellular membranes to release nuclear precursor mRNAs. This insight could be particularly relevant for studies on gene expression and RNA processing.

The GITC-T method, an adaptation of the TRIzol method, minimizes the amount of TRIzol reagent required for extracting total RNA from animal samples, thereby reducing experimental costs without complicating the extraction process. This approach yields total RNA comparable to that obtained through the TRIzol method, with the added advantages of higher yield and purity. Given these benefits, the cost-effective and efficient GITC-T method emerges as a particularly suitable option for smaller-scale laboratories seeking to maintain high standards of RNA extraction while managing limited resources. The GITC-T method reduces the volume of the TRIzol reagent required, thereby decreasing experimental costs while still obtaining high standards of efficiency and quality of RNA products.

Conclusions

The GITC-T method, an adaptation of the TRIzol method, minimizes the amount of TRIzol reagent required for extracting total RNA from animal samples, thereby reducing experimental costs without complicating the extraction process. This approach yields total RNA comparable to that obtained through the TRIzol method, with the added advantages of higher yield and purity. At present, we have only validated the effectiveness of the GITC-T method on human and mouse samples, and further validation is needed to determine whether it is generalizable to other model and non-model species. Given these benefits, the cost-effective and efficient GITC-T method emerges as a particularly suitable option for smaller-scale laboratories seeking to maintain high standards of RNA extraction while managing limited resources.

Supplemental Information

Supplemental Information 1 Comparison of RNA integrity between different sample types

(A) RNA electrophoresis patterns of different samples extracted by the Trizol and GITC-T methods. (B) The intensity ratio of 28S to 18S in (A) was statistically compared to reflect the difference in the integrity of RNA extracted by the GITC-T and Trizol methods.

Supplemental Information 2 Quantify the transcript abundance of the GAPDH and PIRAT1 genes in total RNA samples extracted using different methods

Based on the q-PCR results from Fig. 5B, statistical analysis of the three sample groups shows that gDNA residues are present in the total RNA extracted by both methods. However, the GITC-T method has relatively less residual gDNA, resulting in little impact on the GAPDH (upper panel) and PIRAT1 (lower panel) genes Ct values between the groups with and without gDNA removal. On the far right is the electrophoresis image of the q-PCR products for the GAPDH and PIRAT1 genes, which visually illustrates the impact of residual gDNA in the total RNA samples.

Supplemental Information 3 Denaturing agarose gel electropherogram of total RNA from animal samples obtained by different extraction methods

Supplemental Information 4 List of reagents used in this study

Supplemental Information 5 List of instruments used in this study

Supplemental Information 6 List of primers for q-PCR

Supplemental Information 7 System of real-time fluorescence quantitative PCR reaction

Supplemental Information 8 Total RNA quantity and purity were extracted from samples of different samples

Supplemental Information 9 Author checklist

Supplemental Information 10 Q-PCR data

Supplemental Information 11 Concentration and purity RNA data

Supplemental Information 12 MIQE checklist

Additional Information and Declarations

Competing Interests

Author Contributions

Human Ethics

Animal Ethics

DNA Deposition

Data Availability

The authors declare there are no competing interests.

Yumei Zeng performed the experiments, prepared figures and/or tables, authored or reviewed drafts of the article, and approved the final draft.

Xiaoxue Tang performed the experiments, prepared figures and/or tables, authored or reviewed drafts of the article, and approved the final draft.

Jinwen Chen analyzed the data, authored or reviewed drafts of the article, and approved the final draft.

Xi Kang analyzed the data, prepared figures and/or tables, and approved the final draft.

Dazhang Bai conceived and designed the experiments, prepared figures and/or tables, authored or reviewed drafts of the article, and approved the final draft.

The following information was supplied relating to ethical approvals (i.e., approving body and any reference numbers):

the Ethics Committee of the Affiliated Hospital of North Sichuan Medical College.

The following information was supplied relating to ethical approvals (i.e., approving body and any reference numbers):

the Animal Research and Ethics Committee of North Sichuan Medical College.

The following information was supplied regarding the deposition of DNA sequences:

The sequences are available at NCBI: PRJNA1111442.

http://www.ncbi.nlm.nih.gov/bioproject/1111442.

The following information was supplied regarding data availability:

The sequences are available at NCBI: PRJNA1111442.

The RNA-seq data are available at figshare: Bai, Dazhang (2024). Optimizing Total RNA Extraction Method for Animal Tissues. figshare. Dataset. https://doi.org/10.6084/m9.figshare.25678368.v1.

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
