# Peer review of "Optimizing total RNA extraction method for human and mice samples"

_PeerJ, doi:10.7717/peerj.18072_

## Round 0.1 · original submission · Major Revisions

Please provide a detailed explanation of the concerns raised by the reviewers. Specifically, it is important to include the DNase treatment of RNA preps, quantification of RNA using a Qubit fluorometer, and quality assessment using a Bioanalyzer. In addition, the use of No-RT controls and appropriate housekeeping genes in RT-PCR is necessary to validate the effectiveness of the methods described.

Reviewer 1 ·

Basic reporting

Zeng and Tang et al., describe the incorporation of guanidine isothiocyanate (GITC) and/ or sodium dodecyl sulfate (SDS) to the current TRIzol method for RNA extraction from human blood and cell lines including mouse brain with quality and quantity assessment using nanodrop, agarose gel electrophoresis, qRT-PCR and RNA-Seq. However, there are several limitations that need to be addressed:
1- TRIzol itself is a solution of guanidium isothiocyanate (GITC) and phenol. This needs to be clearly stated.

2- 1000ul of TRIzol is used in comparison to 800 ul TRIzol + 200ul SDS or 100ul SDS + 100ul GITC. There is saving of only 200ul of TRIzol /sample as the methodology does not eliminate TRIzol. The modified method adds ice bath and 12 min to the modified steps.

3- Trichloromethane is also known as chloroform. Suggest using chloroform to avoid confusion and comparison to already established methods.

Experimental design

4- Residual genomic DNA (gDNA) contaminating RNA samples is a concern in RNA preps and it is necessary to treat all RNA samples with DNase. Steps need to be taken to ensure there is no gDNA contamination.

(a) Nanodrop is a spectrophotometric method of RNA quality (contaminants such as GITC or phenol) and quantity assessment. However, nanodrop usually overestimates concentration considerably and is not specific to either DNA or RNA. Qubit fluorometer is the gold standard for estimation of nucleic acids and is specific to DNA or RNA. Nanodrop also does not determine nucleic acid degradation.
(b) Non-specific amplification due to gDNA contamination will overestimate the abundance of transcript levels and can affect the RT-qPCR results. gDNA can be detected in qRT-PCR reactions by using primer pairs annealing to intergenic regions or an intron of the gene of interest. A no reverse transcriptase control (No RT) control that carries out the reverse transcription step of a qRT-PCR experiment in the absence of reverse transcriptase. This control assesses the amount of DNA contamination present in an RNA preparation.
(c) Current standards for NGS require estimation of RNA integrity number (RIN) using a Bioanalyzer. RINs are used as a cutoff for acceptance or rejection of sample for RNA -Seq. The authors have shown a plot like ‘Bioanalyzer’ output and mention ‘quality control testing provided by a biotechnology company’. However, no accompanying gel images were provided. Please state the methodology used for this assessment.
(d) Agarose gel electrophoresis was used to check RNA quality. While it is not necessary to run a denaturing gel just to check RNA quality it is suggested because most RNA forms extensive secondary structure via intramolecular base pairing, and this prevents it from migrating strictly according to its size. The gel images shown are not high resolution with plots showing intensity of the 28s and 18s bands. A single analysis using Bioanalyzer will provide RINs, provide plots, and gel images with resolution to show gDNA contamination, if any.
(e) Normally the Ct values of GAPDH is between 18-22. Ct value of housekeeping GAPDH used as control is ~35 cycles and the test genes is ~29 cycles.

5- In a clinical research setting involving human subjects, tissues and blood are not immediately processed, but either snap frozen in LN2 or blood collected in tube containing RNA stabilizers (for e.g., PAXgene tubes). This limits the general use of the current protocol. (EDTA)-coated tubes are the conventional containers for blood collection that do not contain RNA stabilizing chemicals to protect RNA from degradation or induce or repress gene expression without a preservative.

6- Is the RNA extraction rate (ng/mg) the amount of total RNA obtained per mg of tissue. How was this determined for blood?

Validity of the findings

Overall, DNase treatment of RNA preps with quantification of RNA using Qubit fluorometer and quality assessment with Bioanalyzer with use of No-RT controls and appropriate housekeeping genes in RT-PCR is needed to validate the efficacy of the methods described.

Additional comments

Minor suggestions:
Table 1: Add ‘water to ‘20ul DEPC’ for dissolution of RNA
Abstract: Replace Microarray ‘DNA’ Hybridization to ‘expression’ microarrays where cRNA is hybridized

Annotated reviews are not available for download in order to protect the identity of reviewers who chose to remain anonymous.

Reviewer 2 ·

Basic reporting

This manuscript presents an investigation where the TRIZol RNA isolation method was optimized by adding GITC and SDS in the cell lysis step. The results are valuable since the isolated RNA presents high integrity and purity and is suitable for qPCR and RNASeq assays. However, the manuscript needs to be improved before being accepted for publication.

Title
• When I read the title, I expected that the authors would have isolated RNA from the tissues of different types of animals. Therefore, in my opinion, it is better to specify (in the title) that it was an optimization of RNA isolation for human and mice samples rather than for “animal tissues”.

Abstract:
• The results presented are very general. Therefore, continuous and/or percentage data should be included. For example, how many nanograms of RNA were obtained in the modified protocol compared to the traditional one? How was the variation in purity and integrity (RIN, purity ratios)? How does the CT number change between one protocol and the other? What about the increase (folds) in the number of transcripts?

Introduction:
• This section mentions that there are already published manuscripts in which RNA isolation with TRIzol has been optimized (Lines 61-62). In that sense, the authors could justify the need for further optimization of the TRIzol method by highlighting the gaps in current knowledge.

Discussion:
The discussion is concise and mentions the challenges the authors encountered while developing their methodology. However, there are no citations in most of the paragraphs of this section. In this sense, the discussion could be enriched by addressing issues such as the relevance of the lysis step in RNA extraction, the similarities and differences observed with other optimizations of the TRIzol method, the advantage in reducing the volume of TRIzol used, etc.

Figures and tables
Based on the results presented in Figures 1-6 and Tables 2-4, if the authors have four sample types (brain cortex tissue, blood and U87-MG and Hela S3 cells), why did they use different sample types for each analysis? For example:
A. Figure 1 (RNA concentration, integrity, and intensity of 28S and 18S bands): the sample type used is not specified. Also, there is no TRIzol isolation data (1A-F).
B. Figure 2 (purity ratios): the sample type used is not specified.
C. Figure 3 (bioanalyzer results): U87-MG cells were used.
D. Figure 4 (RNASeq): authors used blood and U87-MG and Hela S3 cells.
E. Figure 5 (qPCR): the sample type used is not specified.
Why is there no consistency in the type of tissue used for each analysis? What were the criteria used to determine which sample type to use for each analysis?
Figures 1E-F: How was the intensity of the bands measured? It is not mentioned in the methodology.

Experimental design

• If the authors intend to provide a universally applicable method for total RNA extraction from animal samples (lines 82-83), why only test it on two species?
• If the primary aim was to reduce the volume of the TRIzol reagent required (line 80), why didn't the authors try to extract RNA using a smaller volume of TRIzol?
• The authors said: “The abundance of the target gene transcripts in the RNA samples was determined by analyzing the threshold cycle number (Ct) from the q-PCR reactions, allowing for the estimation of the starting copy number of the transcripts in the total RNA samples.”. How was the number of initial copies estimated if no absolute quantification methodology was described?
• “Concentration gradients of 3 mol/L (M), 4 M, and 5 M for GITC and 5%, 10%, and 15% for SDS were tested to determine the optimal concentrations.” – in the methodology, the authors make no mention of these gradients.
• The methodology does not mention how the analysis to determine the “RNA sequence compositions” (line 332) was performed. What kind of data was generated with RNASeq?

Validity of the findings

• Before proceeding with cDNA synthesis, RNA extractions are usually treated with DNase. When is or is not DNase treatment necessary?
• Was the distribution and homogeneity of variances analyzed before using parametric methods? (line 277)
• In humans and mice, 28S has a size of 4000-5000 bp and 18S 2000 bp, whereas in the gel presented by the authors (Figs. 1C-D) both bands show sizes below 1500 bp, how do the authors explain these differences in the expected band sizes? What molecular weight marker did the authors use?
• Regarding the amplification of GAPDH using qPCR, how did authors determine which of the two peaks observed corresponds to the expected qPCR product? Did the authors use a non-template control? How do authors know that one of the peaks is a primer dimer and not another nonspecific product type? Primer dimers typically have a Tm < 80 °C.

Additional comments

Minor comments:
Line 20: CTAB is a common abbreviation: however, I suggest writing the full name in the abstract.
Line 25-26: the aim of the manuscript (“The objective is to optimize RNA extraction from animal tissues”) should appear at the end of the background, not in the methods section.
Line 26: specify the types of animal tissues used in the study.
Line 32-33: “and greater transcript abundance of total RNA” - is this the result of the qPCR or RNA-Seq assay? From how it is written, it seems this type of result is obtained by qPCR and not by RNASeq.
Line 52: TRIzol is not a kit, it is a reagent. Kits usually include cell lysis buffer, elution buffer, etc.
Line 110: “Thermo Fisher Scientific” instead of “Thermo”.
Line 113: use “q-PCR” or “Q-PCR” throughout the document.
Line 152: “Total RNA was extracted from various sample types”. Please, specify the types of animal samples used.
Line 153 & 158: there are some differences between the guidelines of the manufacturer and the procedure followed by the authors (for example, incubation overnight, the time of centrifugation of the sample+IPA, etc.). So, it is necessary to say that some modifications were made to the manufacturer protocol.
Line 158: although trichloromethane is correct, the scientific community uses the term chloroform more. So, I suggest using the term chloroform.
Lines 171 and 189: in all three extraction methods, the separation, RNA precipitation, washing and drying, and reconstitution steps are the same; so, these steps could be omitted in the last two methods (mention that they are the same as in the TRIzol method).
Line 211: “Total RNA Yield Assay involved measuring the concentration of total RNA extracted from animal samples using various methods with Thermo’s NanoDrop One Microvolume UV-Vis Spectrophotometer”. Which were those various methods?
Lines 213-214: these two lines could be omitted.
Line 222: the information in this subsection could be integrated with the "Total RNA yield assay" section.
Line 236: the information on the agarose gel preparation could be omitted as well as the information on gel casting (except for the percentage of agarose gel and the dye used).
Line 260: what is the expected size of the amplicon?
Line 269: what was the temperature range used for the dissociation step?
Line 272: what is the difference between transcriptome and whole transcriptome sequencing? There is no information about the platform on which the RNA sequencing was done, how the libraries were prepared, about replicates, etc.
Line 276: could you provide more information on the experimental design mentioned?
Line 292: There are no boxplots in figures 1A-B, figures 1A-B are bar plots.
Line 295: There are no boxplots in figures 1E-F, figures 1A-B are bar plots.
Line 303-307: I suggest adding a citation.
Line 319-321: “Furthermore, the integrity assessment extended to U87-MG cell samples processed by both methods, with sequencing pre-tests conducted by Wuhan Gene Read Biotechnology Co., Ltd.”. The idea seems incomplete, I suggest rewriting the statement to make it more understandable.
Line 341: how many nanograms of cDNA (equivalent RNA) were used in each qPCR reaction?
Line 368: the year of publication is missing in the citation.
Lines 370-372: I suggest adding citations.
Line 396-397: in what order do the authors suggest they should be added?
Line 426: what about the toxicity of GITC, is it lower than the toxicity of Trizol?
Line 431: the conclusion could be enriched by adding information on the limitations and scope for further research. For example, could this new protocol be optimized in any way? Would it be worth using for RNA extraction in non-model species?
Figure 3A – “Results of RNA sample integrity quality-control testing provided by a biotechnology company”. Is this the result of a Bioanalyzer?

Reviewer 3 ·

Basic reporting

No comments

Experimental design

The work done by authors is good. However, the reason for optimizing the method is not appropriate as guanidine isothiocyanate is also expensive reagent and when used with trizole will increase thecost of the reaction.

Validity of the findings

no comment

Additional comments

no comments

---

## Round 0.2 · Minor Revisions

Please address the reviewer's final comments and submit a revised version along with a rebuttal letter.

Reviewer 2 ·

Basic reporting

The authors have improved the quality of the manuscript by taking into account most of the reviewers' suggestions, even repeating some of the experimental work. I have a few more recommendations:

Line 233: Specify the brand of DNase used and the amount of RNA that was retrotranscribed (1000 ng according to the response letter).

Line 240: regarding "the temperature range used for the dissociation step"; I referred to the melt curve analysis. As far as I can see in Figure 5A, it was 65-95 °C (it can be mentioned in the methodology as https://peerj.com/articles/1292/ did).

Lines 241-243: from the explanation made by the authors in the response letter, I understand that they did relative quantitative PCR. I do not know this methodology or whether it is a standard or valid practice in qPCR to estimate the number of initial copies of a transcript. Then, I consider that a citation should be placed mentioning that the estimation was done according to the method of X author.

Line 339: I recommend adding in the supplementary table 4 that 1 ul of cDNA was used and that it is equivalent to 17 ng total RNA.

Line 365: In the abstract, the authors use the name "hexadecyl trimethyl ammonium bromide" instead of “cetyl trimethyl ammonium bromide”. Both terms are valid; however, I suggest using only one of them to avoid confusion.

Experimental design

'no comment'

Validity of the findings

'no comment'

Additional comments

'no comment'

---

## Round 0.3 · Minor Revisions

Please consider the following editorial and style suggestions in the attached PDF file, and return a revised version at your convenience.

---

## Round 0.4 · accepted · Accept

Thank you for addressing the comments and suggestions. Your manuscript has significantly improved.